# Effects of Modifications on the Immunosuppressive Properties of Cyclolinopeptide A and Its Analogs in Animal Experimental Models

**DOI:** 10.3390/molecules26092538

**Published:** 2021-04-27

**Authors:** Michał Zimecki, Krzysztof Kaczmarek

**Affiliations:** 1Laboratory of Immunobiology, Hirszfeld Institute of Immunology and Experimental Therapy, Polish Academy of Sciences, R. Weigla Str. 12, 53-114 Wrocław, Poland; 2Institute of Organic Chemistry, Lodz University of Technology, S. Żeromskiego Str. 116, 90-924 Łódź, Poland

**Keywords:** cyclolinopeptide A (CLA), cyclosporine A, immunosuppression, PGE2, edge-to-face, *cis*-peptide bond, homo amino acids

## Abstract

The consequences of manipulations in structure and amino acid composition of native cyclolinopeptide A (CLA) from linen seeds, and its linear precursor on their biological activities and mechanisms of action, are reviewed. The modifications included truncation of the peptide chain, replacement of amino acid residues with proteinogenic or non-proteinogenic ones, modifications of peptide bond, and others. The studies revealed changes in the immunosuppressive potency of these analogs investigated in a number of in vitro and in vivo experimental models, predominantly in rodents, as well as differences in their postulated mechanism of action. The modified peptides were compared with cyclosporine A and parent CLA. Some of the synthesized and investigated peptides show potential therapeutic usefulness.

## 1. Introduction

A cyclic nonapeptide Cyclolinopeptide A (CLA) was isolated from linen seeds oil [1]. The structure of CLA, cyclo-(Leu^1^-Ile^2^-Ile^3^-Leu^4^-Val^5^-Pro^6^-Pro^7^-Phe^8^-Phe^9^-), is presented in Figure 1. 

CLA belongs to the plant *Orbitides* family and was recently more closely characterized [2,3]. Although flaxseed extracts contain a variety of pharmacologically active compounds [2,3], the biological properties of CLA and its analogs have been investigated most extensively. In these studies, the native peptide was subjected to several modifications in order to improve its immunosuppressive efficacy and solubility, and to identify its key sequences in the structure. The modifications included, among others, truncation of peptide chain for both cyclic and linear analogs, as well as substitution of particular amino acid residues. The modifications paralleled evaluation of their immunosuppressive activities and mechanism of action.

## 2. The Effects of CLA on the Humoral and Cellular Immune Responses

Immunosuppressive properties of CLA were described in several models 30 years ago [4]. The peptide suppressed the humoral and cellular immune response to sheep erythrocytes (SRBC), mitogen induced proliferation of human peripheral blood mononuclear cells (PBMC) and activity of IL-1 and IL-2. Other activities of CLA included suppression of graft-versus host reaction, allogeneic graft rejection, post adjuvant polyarthritis in rats and retard of hemolytic anemia in New Zealand Black mice. CLA was effective when administered by various routs, i.e., intravenously, intraperitoneally or per os, and showed the suppressive potency comparable to that of cyclosporine A (CsA). The authors underline that efficacy of the peptide by oral administration may have potential therapeutic value. In a series of CLA linear analogs the peptides were successively truncated from the N-terminus and their immunosuppressive activity were evaluated [5]. The immunosuppressive potency of the peptides gradually decreased with shortening the peptide chain. Interestingly, the octapeptide with the sequence H-Ile-Ile-Leu-Val-Pro-Pro-Phe-Phe-OH appeared to be most suppressive in the models of humoral and cellular immune response, and inhibition of lipopolysaccharide (LPS) induced interleukin 1 and 6 and tumor necrosis alpha (TNF α) production played a major role in its activity.

## 3. Role of Single Amino Acids in the Activity of CLA Analogs

In subsequent investigations the importance of single amino acid residue on the activity of the peptide was determined by replacements with another amino acid residue. In five linear and five cyclic peptides, leucine, isoleucine and valine residues were consecutively replaced by threonine and their immunosuppressive activities were tested for the suppression of the humoral and cellular response to SRBC [6]. It appeared that the inhibitory actions of the peptides in the humoral response, comparable to that of CsA, were observed with linear peptides bearing the sequences H-Leu-Ile-Thr^3^-Leu-Val-Pro-Pro-Phe-Phe-OH and H-Leu-Ile-Ile-Leu-Thr^5^-Pro-Pro-Phe-Phe-OH. On the other hand, the suppression stronger than that exerted by CsA was registered for the peptide H-Thr^1^-Ile-Ile-Leu-Thr^5^-Pro-Pro-Phe-Phe-OH. In another study, a significant suppression of both types of the immune response was achieved by substitution of one or both phenylalanine residues with tyrosine [7]. The replacement of both phenylalanine residues with tyrosine was most effective, particularly in the cyclic form of the peptide c (Leu-Ile-Ile-Leu-Val-Pro-Pro-Tyr-Tyr-) and exceeded the activities of CsA and parent CLA. On the other hand, the consecutive substitution of each following amino acid residue with alanine residue (so called “alanine scan”) did not reveal any particular peptide of a special immunosuppressive potency [8]. 

All these analogs with single alanine residue were less effective in the humoral immune response, as compared with parent CLA, but showed a comparable efficacy with CLA in the suppression of the cellular response. Interestingly, the suppressive property of linear peptide H-Gly-Ile-Ile-Leu-Val-Pro-Pro-Phe-Phe-OH [9] could be further increased by additional glycine residues from both termini of the peptide. 

## 4. Effects of Modified Amino Acids on Activity and Establishment of a Key Pro-Pro-Phe-Phe Sequence in CLA Analogs

A study on immunosuppressive activity of linear and cyclic analogs of the peptides, in which one or both phenylalanine residues were substituted by their sulfonated derivatives [10], revealed no differences between the actions of CsA and these compounds in the model of the humoral immune response to SRBC. However, in the model of the cellular immune response to SRBC, a linear peptide with sulfonated phenylalanine in position nine was more suppressive than CsA. Another modification involved the synthesis of linear and cyclic analogues of CLA with two dipeptide segments, Val^5^-Pro^6^ and Pro^6^-Pro^7^, replaced by their tetrazole derivatives [11] and the evaluation of their immunosuppressive activities in the humoral response test. The results showed that, even at low doses, the immunosuppressive activity of the cyclic analogs was equal to the activity exhibited by CsA and native CLA. The conformational data, in association with biological results, indicate that the Pro-Pro-Phe-Phe sequence and the preservation of the CLA backbone conformation are very important for the immunosuppressive activity. In another study linear and cyclic CLA analogues containing *α*-hydroxymethylleucine (HmL) in position 1 or 4, or *α*-hydroxymethylvaline (HmV) in position five were synthesized and subjected to cyclization [12]. Although the peptide containing HmL in position 4 was by 25% less suppressive in the lymphocyte proliferation assay than parent CLA, a clear advantage of such modification was that its solubility in water surpassed four times that of CLA. 

## 5. Significance of the Spatial Conformation in the Activity of CLA

To evaluate the role and significance of “edge-to-face” interaction in the process of molecular recognition by receptors, three linear precursors and three cyclic analogues of CLA were synthesized with one or both Phe^8,9^ residues replaced by *β*^3^-homophenylalanine (*β*^3^hPhe) residue [13]. The conformational analysis by NMR was carried out on the CLA analogue, in which Phe^8^ was substituted by *β*^3^hPhe residue to study the influence of the mutation on the three-dimensional structure. These studies revealed that, when compared to the CLA structure, a preservation of the *cis* geometry of Pro-Pro amide bond and the stacking of the rings of Pro^7^-*β*^3^hPhe^8^. On the other hand, they found a different structure of Phe^9^ side chain that is located on the opposite side of the peptide plane. Such behavior excludes the existence of the face-to-edge interaction between phenyl rings, as it occurs in CLA. The peptides were tested in the models of humoral immune response to SRBC and cellular response to ovalbumin (OVA). The best immunosuppressive activities were registered in the case of the linear peptide and its cyclic counterpart modified with *β*^3^hPhe in position 8. The authors concluded that the postulated edge-to-face interaction of phenylalanine rings is not critical for immunosuppressive effects. 

## 6. Modifications of CLA Structure by Unnatural Aminoacids

CLA analogs were also modified with (*S*)-*β^2^*-isoproline or (*S*)-*β^3^*-homoproline in position 7, respectively [14]. Peptides 1–6 existed as a mixture of four isomers due to *cis/trans* isomerization of the Xxx-Pro peptide bond. The major isomers of peptides 1, 3, and 4 contained all peptide bonds of the *trans* geometry. The geometry of the proline-proline bond of the second populated isomer of peptides 3 and 4 was *cis*. The inhibition of mitogen-induced splenocyte proliferation and the secondary humoral immune response to SRBC by the peptides strongly depended on position and content of proline isomers. Of importance, the peptides were devoid of cell toxicity, even at 100 µg/ml concentration. It seems that the replacement of Pro by any of *β*-Pro residues may be useful for modification of CLA suppressive activity and for lowering its cytotoxicity. An exceptionally strong immunosuppressor was also obtained by modification of CLA in position 8 with homophenylalanine [15]. The peptide completely blocked proliferation of mouse splenocytes to mitogens, albeit only at 100 µg/mL concentration, thus being much less potent in comparison with CsA used as a reference drug.

Very promising suppressive analogs were also obtained by modification of linear and cyclic CLA with *S* or *R*-*γ^3^*-bis(homophenylalanine) in positions 8 or 9, or both [16]. Only cyclic peptides inhibited the proliferative response of PBMC to phytohemagglutinin (PHA) and exhibited a dose dependent toxicity except a peptide substituted with *S*-*ɣ^3^*hhPhe^9^. On the other hand, a linear peptide containing—(*R*)-*ɣ^3^*hhPhe^8^-(*R*)-*ɣ*^3^hhPhe^9^- very strongly inhibited LPS-induced TNF-α production by human whole blood cultures. Strong suppressive actions of both peptides were confirmed in inductive and effectual phases of the delayed type hypersensitivity, as well as in the carrageenan-induced foot pad edema, where the cyclic peptide was more potent. In a similar study [17], CLA was modified with (*S)-* or *(R*)-*γ^4^*-bis(homophenylalanine) in positions 8 and/or 9. These modifications changed the flexibility of the analogues and distribution of the intramolecular hydrogen bonds. The proliferative response of PMBC to PHA was registered only with cyclic peptides, as in the preceding article, but LPS-induced TNF-α production in whole blood cell cultures was inhibited by both linear and cyclic peptides.

In a last attempt to modify the CLA structure [18], the synthesis of new analogues of cyclolinopeptide A and their linear precursors, modified with (*R*)- and/or (*S*)-4-methylpseudoproline in the Pro^6^-Pro^7^ fragment, was performed. The structure of the peptides was also characterized by application of nuclear magnetic resonance (NMR) and circular dichroism (CD). The results showed that only peptides 7 and 8 modified with (*R*)-4-methylpseudoproline residue (*(R)*-(αMe)Ser(ΨPro) in the position 6, or 7, respectively, strongly suppressed mitogen-induced splenocyte proliferation and the humoral immune response, with peptide **8** being more potent. In addition, peptide 8 more strongly elevated expression of Fas, a proapoptotic signaling molecule in Jurkat T cells. The authors postulated that the more potent activity of peptide 8, compared to the parent molecule and other studied peptides, was due to its more flexible structure since CD and NMR spectra showed that replacement of Pro^6^ by *(R)*-(αMe)Ser(ΨPro) caused much deeper conformational changes than the analogous modification of the Pro^7^ residue. A possible consequence of such increased conformational freedom would be a better accommodation to its putative cellular receptor. The summary of structure/activity relationships of CLA analogs is presented below.

Summary of structure/activity relationships in modified CLA analogs:absence of Leu^1^ is not important for CLA activityPro^6^-Pro^7^-Phe^8^-Phe^9^ sequence is essential for the immunosuppressive activity in native CLALeu, Ileu and Val may be replaced by Thr without loss of CLA activityPhe^8^ or both Phe^8^ and Phe^9^ may be replaced by Tyr without loss of CLA activitysulfonated Phe or tetrazole derivatives of Val^5^-Pro^6^ and Pro^6^-Pro^7^ do not alter CLA activitysubstitution with hydroxymethylleucine at position 1 or 4 or with hydroxymethylvaline^4^ do not significantly affect suppressive activity of CLAedge-to-face interaction of the phenylalanine rings is not critical for CLA activityCLA modified with *β^3^*-homophenylalanine^8^ enhances its anti-proliferative actioncyclic peptides containing *(S)* or *(R)*-*ɣ*^3^-bis-homophenylalanine^8^ are strongly anti-proliferative but linear ones are inhibitory only in LPS-induced TNF α productionanti-proliferative actions of CLA analogs modified with *(R)*- or *(S)*-4-methylpseudoproline at position 6 or 7 are enhanced and associated with proapoptotic activitythe cyclic tetrapeptide loses the anti-proliferative activity, but acquires strong anti-inflammatory property associated with its regulatory effect on prostanoid metabolism

## 7. Modification of Suppressive Action of Methotrexate by CLA Analogs

A possibility of modification of the suppressive action of methotrexate (MTX) by CLA and its linear analogs in in vitro humoral immune response to SRBC was also investigated [19]. CLA, at a low dose (0.1µg/mL), deepened inhibition of antibody forming cell number when combined with MTX (0.25–0.5 mM), showing an additive effect, potentially of clinical importance. Although the modified CLA structures and their linear precursors did not show any significant effect on the immune response, they exhibited antagonistic action with respect to MTX.

## 8. Activity of c(Pro-Pro-*β*^3^hPhe-Phe-), the Cyclic Tetrapeptide

Perhaps a most advanced modification of CLA structure was received through cyclization of analogs of its tetrapeptide fragment Pro-Pro-Phe-Phe containing unnatural amino acid *β*^3^-homophenylalanine (*β*^3^hPhe) [20]. The structure of the tetrapeptide, termed 4B8M, is presented in Figure 2.

The tetrapeptide c(Pro-Pro-*β*^3^hPhe-Phe-) was investigated in several in vitro and in vivo experimental models using intraperitoneal, per os and topical routs of administration. It appeared that its toxic action towards mouse splenocytes was much lower when compared to parent CLA and cyclosporine (CsA). The peptide exhibited strong inhibitory action in the model of in vitro secondary immune response to SRBC. However, its ability to inhibit mitogen-induced lymphocyte proliferation and LPS-elicited TNF-α production was low. On the other hand, the peptide demonstrated remarkably strong anti-inflammatory actions in comparison with reference drugs in the models of contact sensitivity to oxazolone and toluene diisocyanate, nonspecific skin irritations, carrageenan inflammation in air pouch, ovalbumin induced pleurisy and dextran sulfate-induced colitis. The mechanism of action of the cyclic tetrapeptide is described below.

## 9. Other Biological Activities of CLA

Apart from more frequently investigated properties of CLA, such as effects on mitogen-induced lymphocyte proliferation, cytokine production and humoral and cellular immune responses, the peptide exhibited immunosuppressive actions in other models. CLA ameliorated post-adjuvant polyarthritis in rats and delayed development of hemolytic autoimmune anemia in NZB mice [4]. Inhibition of autoimmune response was also found in our study on collagen induced arthritis in DBA1 mice (unpublished). CLA, with a series of its analogs, inhibited also growth of human *Plasmodium falciparum* in culture [21]. The inhibitory action of the peptides was not likely associated with their putative calcineurin receptor on cell membrane. Interestingly, the antimalarial action was somehow linked to the hydrophobic residue of the peptides. The cyclolinopeptides, including CLA, inhibited osteoclast differentiation [22]. Such property could be of advantage in delaying progress of osteoporosis. In addition, CLA was shown to inhibit growth of bone giant cell tumor by arresting cells in G0/G1 phase of cell cycle [23]. Interestingly, our data from an investigation on 13-week toxicity in rats (unpublished) revealed that CLA lowered serum cholesterol level, albeit only in females. The immunosuppressive actions of CLA and some of its analogs are presented in Table 1.

## 10. The Mechanism of Action

Due to the cyclic structure and close similarity of CLA actions to the immunosuppressive activity of calcineurin inhibitors, such as cyclosporine A (CsA) and tacrolimus, a possibility existed that both classes of compounds share the same mechanism of action. Evidence was delivered that CLA and its analog bound bovine cyclophilin [24]. Nevertheless, these interactions appeared to be many folds lower in comparison with CsA. The authors also pointed at a crucial role of Pro^6^ residue in this interaction [25]. Another group of researchers [26] confirmed that this type of interaction is weak since the inhibition of calcineurin activity, important in T cell signaling and proliferation [27] required 10× higher concentration of CLA than for CsA or tacrolimus. Strong inhibition of mitogen induced proliferation of PBMC by CLA analogs was associated with significant changes in expression of cell signaling molecules in Jurkat T cells [17,18], such as block of caspase 3 responsible for activation of T cells and IL-2 production [28] and increase of Fas [16] involved in apoptosis [29]. In turn, the mechanism of action of cyclic tetrapeptide cyclo-(Pro-Pro-*β*^3^hPhe-Phe-) was clearly associated with its effects on prostanoid activity [20]. The peptide lowered LPS-induced expression of EP1 and EP3 but not EP2 and EP4 receptors on Jurkat cells. Such a differential action of the peptide on prostaglandin E2 (PGE2) receptors explains its anti-inflammatory properties since PGE2 exerts proinflammatory actions via EP1 and EP3 and anti-inflammatory ones using EP2 and EP4 receptors [30]. In addition, the tetrapeptide lowered expression of adhesion molecule ICAM-1 and induced expression of COX-2 on keratinocytes. These events correlate well with the peptide’s actions, particularly in skin inflammation, since COX-2 induced by the peptide in keratinocytes [20] is responsible for lowering ICAM-1 expression [31] and chemokine production [32] by keratinocytes.

## 11. Conclusions

The modifications in structure and amino acid sequence of CLA and its analogs lead to differential consequences in their immunosuppressive activities and mechanism of action. In general, linear peptides are less potent than cyclic peptides and cyclic peptides more toxic towards lymphocytes. Pro-Pro-Phe-Phe sequence in parent CLA and its analogs plays a key role in the biological activity. Parent CLA exhibits a component of calcineurin inhibition activity and is strongly antiproliferative. This finding is supported by the similarity in CLA action and its cyclic analogs to the action of CsA, particularly if administered to mice during the inductive phase of the immune response. The mechanism of action of CLA analogs may also include arrest in cell cycle. Nevertheless, the successive truncation of the peptide chain results in a gradual loss of the suppressive property. Although the cyclic tetrapeptide still contains the key amino acid sequence, it loses the antiproliferative activity and becomes predominantly anti-inflammatory with the mechanism of action associated with prostanoid metabolism. In conclusion, CLA analogs may be of potential therapeutic value due to better solubility and lower toxicity in comparison to CsA and parent CLA. A possibility also exist that additive suppressive actions will allow to lower cytotoxic effect of chemotherapeutics at combined application with CLA analogs. It seems that CLA analogs may predominantly have therapeutic potential in the amelioration of autoimmune and inflammatory diseases, and at least in part, involved in the replacement application of classical calcineurin inhibitors, steroidal and nonsteroidal drugs. Future efforts will be focused at further modifications of the cyclic tetrapeptide structure, since this peptide is best characterized in terms of the therapeutic actions in the mouse models and its mechanism of action. In addition, it is low-toxic, and as a small molecule, has a better ability to permeate epithelial tissues. The therapeutic efficacy of the tetrapeptide applied orally may be further enhanced by its binding to carbohydrate or protein carriers.

## Figures and Tables

**Figure 1 molecules-26-02538-f001:**
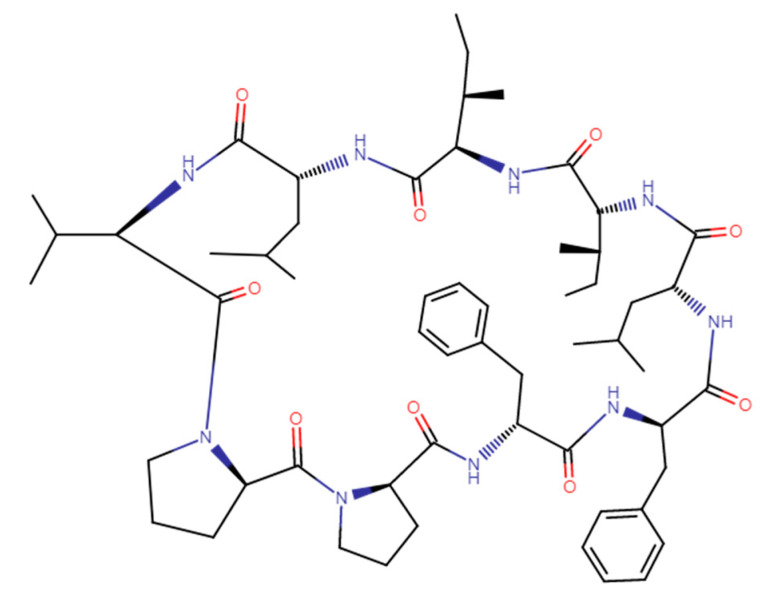
The structure of Cyclolinopeptide A (CAS 33302-55-5).

**Figure 2 molecules-26-02538-f002:**
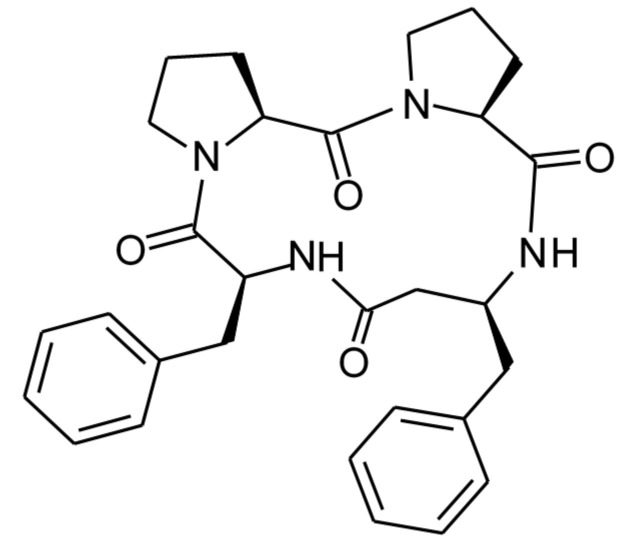
The structure of the cyclic tetrapeptide, cyclo-(L-Pro-L-Pro-L-*β^3^*hPhe-L-Phe-). (*3S,9S,13S,16S*)-9,13-dibenzyl-1,7,10,14-tetraazatricyclo[14.3.0.0^3,7^]nonadecane- 2,8,11,15-tetrone (IUPAC name).

**Table 1 molecules-26-02538-t001:** The immunological activities of cyclolinopeptide A and its analogs.

Peptide	Action	Ref.
cyclolinopeptide A	suppression of: graft-versus-host reaction, allogeneic graft rejection, post-adjuvant polyarthritis, humoral and cellular immune response and reduction of hemolytic anemia in New Zealand Black mice	[4]
cyclolinopeptide A	inhibition of human *Plasmodium falciparum* growth in culture	[21]
cyclolinopeptide A	inhibition of osteoclast differentiation	[22]
cyclolinopeptide A	inhibition of bone giant cell tumor growth	[23]
H-Ile-Ile-Leu-Val-Pro-Pro- *(R)*-*ɣ*^3^hhPhe-(*R)*-*ɣ*^3^hhPhe-Leu-OH c(Leu-Ile-Ile-Leu-Val-Pro-Pro-Phe*-(S)-ɣ*^3^hhPhe-)	suppression of: carrageenan-induced footpad inflammation, LPS-induced TNF α production in whole blood cell cultures and effectual phase of the cellular immune response	[16]
c(Pro-Pro-*β*^3^hPhe-Phe-)	suppression of: contact sensitivity to oxazolone, toluene diisocyanate, nonspecific skin irritation, carrageenan inflammation in air poach, ovalbumin-induced pleurisy and dextran sulfate-induced colitis	[20]

## Data Availability

LUT (Institute of Organic Chemistry), IITD PAN (Laboratory of Immunobiology).

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
