# Peer review of "Effects of Modifications on the Immunosuppressive Properties of Cyclolinopeptide A and Its Analogs in Animal Experimental Models"

_molecules, 2021, doi:10.3390/molecules26092538_

Round 1
Reviewer 1 Report
This paper reviews the effects of modifications on the immunosuppressive ability of CLA and its analogs as well as a brief introduction of the underlying mechanism. The paper is well written except the following comments to be improved before possible publication.
1. The paper covers a lot of research progress related to CLA and the cellular immune response, but the whole article is not easy to read. This is mainly because the layout of the article is not systematic. The author talked about the effects on immune responses in Section 2, which contains a lot of achievements one after another. It should be further divided into small parts (subsections). The same occurs to Section 3.
2. Similarly, for the problem that the article is not easy to read, the author should give some diagrams of the key research progress. This will help the readers for a better understanding.
3. In the conclusion section, the author should discuss the current difficulties and the possible future development direction in this field. Especially because half of the references cited are 20 years ago, this discussion seems particularly necessary.
Reviewer 2 Report
This manuscript gives a brief summary on the reported immunosuppressive activities of cyclolinopeptide A and its analogs. This review paper is interesting and has merits for Molecules. It is recommended for acceptance after major revision with the following concerns addressed.
- In Figure 1, the chemical structure of cyclolinopeptide A is wrong. Each amino reside should has only one chiral center.
- It is recommended to go through the manuscript and improve its fluency. For example:
(1). Line 35, it is unclear the meaning of “a narrow group of researchers”. Does the author mean cyclolinopeptide A is not worth to study? So, what’s the point of this review?
(2). From line 47 to 49, it is unclear the meaning and usefulness of “CLA was effective when administered by various routs, i.e., intravenously, intraperitoneally or per os,”
(3). From line 52 to 54, “Interestingly, the octapeptide with the sequence H-Ile-Ile-Leu-Val-Pro-Pro-Phe-Phe-OH appeared to be the most active.” It is a uncomplete sentence.
(4). Line 71, it is unclear the meaning of “All these analogs with single alanine residue…”
….
- This manuscript is poorly presented. In particular, the immunosuppressive activities of cyclolinopeptide A and its analogs. None of the exact experiential data was disclosed. Without concentration and inhibitory rate, and only using “more suppressive than CsA., less effective, comparable to that of cyclosporine A (CsA), …” are not proper.
- Most of references are too old, the author should cite recent reported papers and give a introduction of new progress.
Round 2
Reviewer 1 Report
The manuscript has been improved and now it might be accepted for publication on Molecules.
Reviewer 2 Report
All my concerns have been well addressed, the manuscript can be published as it is.